# MultiPersona-Align: Zero-Shot Multi-Subject Personalized Image Generation with Layout-Guidance via Dual Representation Alignment

**Veddhanth Chakravarthy**
Shiv Nadar University Chennai
veddhanth@gmail.com

**Samir Kumar Das Mohapatra**
Adobe Systems India
Bangalore
dasmohap@adobe.com

**Chandrakala Shanmuganathan**
Shiv Nadar University Chennai
chandrakalas@snuchennai.edu.in

## Abstract

We propose MultiPersona-Align, a novel approach for personalized image generation that enhances multi-subject diffusion models through self-supervised feature alignment. While existing methods rely primarily on spatial masking for subject control, they often produce semantically inconsistent features that fail to preserve subject-specific visual characteristics. Our method introduces Dual Alignment Framework: (1) Spatially-Aligned Subject-Specific Cross-Attention Mechanism that aligns subject-specific diffusion features with corresponding DINOv2 CLS tokens within spatial regions, and (2) Patch-Aligned Self-Attention that ensures global semantic consistency by aligning full-image diffusion features with DINOv2 patch representations. This approach leverages DINOv2's robust semantic understanding without requiring additional training data or annotations. Experiments on multi-subject generation tasks demonstrate that our alignment losses significantly improve subject fidelity and semantic consistency while maintaining spatial control. The method integrates seamlessly into existing architectures, adding minimal computational overhead during training while providing substantial quality improvements in personalized image generation.

## 1 Introduction

The generation of images containing multiple distinct subjects with precise spatial control remains one of the most challenging problems in controllable image synthesis. While recent advances [1], [2] in diffusion models have enabled impressive text-to-image generation capabilities, maintaining semantic consistency and visual fidelity when generating multiple subjects simultaneously presents significant technical hurdles. Current approaches primarily rely on spatial masking mechanisms that often constrain cross-attention computations to specific image regions, but these methods often suffer from semantic drift and fail to preserve subject-specific visual characteristics across different spatial contexts. The fundamental challenge lies in the tension between spatial control and semantic fidelity. Spatial masking approaches successfully constrain where subjects appear but provide no mechanism to ensure that the generated features within those spatial regions remain semantically consistent with the intended subjects. This limitation becomes particularly pronounced in complex multi-subject scenarios where subjects may interact, overlap, or appear in varying poses and contexts. The resulting generated images often exhibit subject confusion, feature mixing, or semantic inconsistencies that degrade the overall quality and controllability of the synthesis process.

Preprint.

We propose a novel approach that addresses these limitations through dual alignment strategy using self-supervised visual representations from DINOv2. Our approach systematically addresses both local subject consistency and global image coherence, two critical aspects that existing methods fail to handle adequately. The key insight driving our approach is that DINOv2's robust semantic representations, learned through self-supervised training on diverse visual data, can provide strong supervisory signals for guiding diffusion model features toward semantically meaningful and consistent representations. Unlike text-based supervision used in models like CLIP, DINOv2's purely visual training enables it to capture fine-grained semantic relationships without the biases introduced by text-image pairing. This makes it particularly well-suited for spatial control scenarios where precise visual consistency is paramount. Our contributions are: **(1)** We introduce the first systematic approach to representation alignment in multi-subject diffusion models, demonstrating how self-supervised visual representations can guide image generation **(2)** We propose a Dual Alignment Framework that addresses both local-subject specific consistency and global image coherence and **(3)** We propose Spatially-Aligned Subject-Specific Cross-Attention and Patch-Aligned Self-Attention mechanisms.

## 2   Related Work

Early approaches introduced conditioning through text, sketches, or spatial layouts [3], [4]. Foundation models such as Stable Diffusion and DALL·E-2 [5], [3] established text-to-image synthesis via cross-attention, but text alone offers limited spatial control [6]. Recent work shows that aligning representations during training improves efficiency and generation quality [7]. Personalization in diffusion models enables subject-specific generation beyond generic prompts. Early methods like DreamBooth [8] and Textual Inversion [9] preserved identity but suffered from overfitting and poor scalability. Lightweight adapters such as IP-Adapter [10] and FaceAdapter [11] introduced reference-based conditioning, with Subject-Diffusion [12] extending to open-domain personalization. Structural conditioning was pioneered by ControlNet [6] and extended through lighter variants like T2I-Adapter [13], InstructPix2Pix [14], and multi-modal frameworks such as Uni-ControlNet and ControlNet++ [15], [16]. These methods remain largely effective for single-subject scenarios but lack mechanisms for subject-specific alignment. Reference-based approaches including IP-Adapter and FaceAdapter [10], [11] enable identity conditioning yet also focus on single subjects. Multi-subject controllability is addressed by recent methods. MS-Diffusion [17] employs spatial masking, while MultiDiffusion, Composer, and CustomDiffusion [18], [19], [20] support compositionality via attention or fine-tuning. Grounded approaches like GLIGEN and ObjectDiffusion [21], [22] enhance spatial fidelity by coupling prompts with image regions, and Subject-Diffusion [12] enables open-domain personalization. $\lambda$-Eclipse [23] and OmniGen [24] provide multi-subject generation capabilites. Despite these advances, current methods emphasize spatial constraints but lack robust semantic supervision, often leading to inconsistent subject alignment.

Self-supervised learning has transformed visual representation learning by enabling models to learn from large-scale unlabeled data without manual annotations [25], [26]. Foundational frameworks based on pretext tasks and contrastive objectives have advanced the quality and generality of learned features, making them highly effective for downstream tasks such as classification, segmentation, detection, and image generation. The DINO family of models, particularly DINOv2 [27], exemplifies state-of-the-art advances in self-supervised learning. Leveraging momentum contrastive learning and vision transformers, DINOv2 produces representations that are semantically rich, robust, and consistent across diverse viewing conditions. Its high semantic fidelity and spatial granularity make it well-suited for fine-grained supervision in generative modeling, supporting subject-specific and region-aware guidance. Complementing these advances, CLIP [28] has reshaped multimodal representation learning by aligning images and text through large-scale contrastive training. CLIP's joint embeddings enable zero-shot generalization across diverse vision-language tasks, providing a powerful foundation for cross-modal understanding in generative applications. Feature alignment has become a central principle in generative modeling, aiming to bridge the semantic gap between target data distributions and outputs. Early approaches matched global distributions through adversarial training [29] or improved perceptual quality with perceptual losses [30]. Recent work emphasizes multimodal alignment, particularly between visual and textual modalities, to improve controllability and semantic consistency. Large-scale frameworks such as DALL·E 2 and Imagen employ pre-trained embeddings to ground textual prompts in visual space. [7] show that robust alignment of internal features accelerates convergence and enhances fidelity. Guidance strategies, including classifier-free guidance and classifier guidance [31], improve sample quality but are limited by their reliance

on labeled supervision. Self-supervised models such as CLIP have provided scalable semantic supervision without explicit labels. Leveraging these embeddings for conditioning has demonstrated improved alignment, controllability, and diversity in generated samples [32], [33].

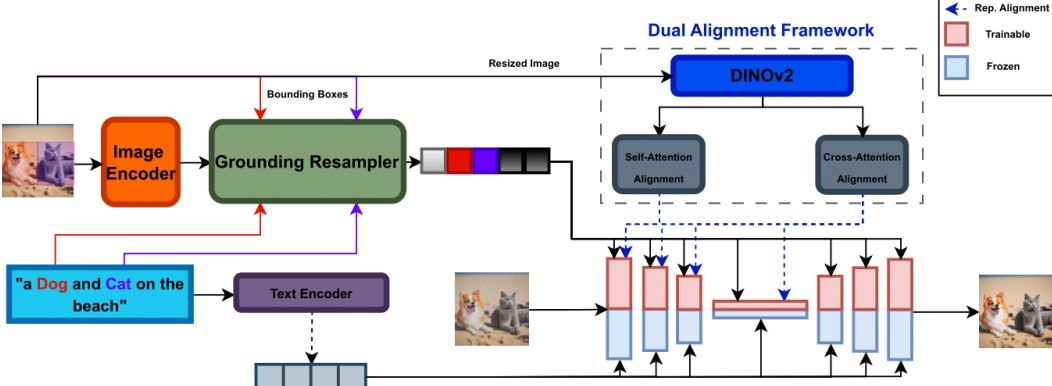

Figure 1: The pipeline processes reference subject images and bounding boxes through a grounding resampler. Our dual alignment framework leverages DINOv2 to provide semantic supervision: (1) Cross-attention alignment matches subject-specific diffusion features with corresponding DINOv2 CLS tokens within spatial regions, and (2) Self-attention alignment ensures global coherence by aligning full-image diffusion features with DINOv2 patch representations. The U-Net denoising process incorporates both alignment pathways while maintaining the frozen base diffusion model.

## 3 Proposed Work

### 3.1 Background: Multi-subject diffusion models

Multi-subject image generation represents one of the most challenging frontiers in diffusion-based synthesis, requiring models to simultaneously preserve individual subject identities while orchestrating coherent spatial compositions. The complexity arises from the need to balance multiple competing objectives: maintaining visual fidelity to reference subjects, respecting spatial layout constraints, ensuring inter-subject consistency, and preserving global image coherence.

MS-Diffusion establishes the current state-of-the-art framework for layout-guided zero-shot multi-subject personalization. The approach introduces two foundational architectural innovations: (1) a grounding resampler, (2) masked cross-attention mechanisms that constrain subject influence to designated spatial regions, preventing unwanted feature bleeding between subjects. The MS-Diffusion framework operates on the standard denoising diffusion objective:

$$\mathcal{L}_{\text{Diffusion}} = \mathbb{E}_{x_0,\epsilon\sim\mathcal{N}(0,I),t,c}\left[\|\epsilon - \epsilon_\theta(x_t,t,c,I_{\text{ref}},B)\|^2\right] \quad (1)$$

where $x_t$ represents noisy latents at timestep $t$, $c$ denotes text conditioning, $\mathbf{I}_{ref} = I_1, I_2, ..., I_K$ are reference subject images, and $\mathbf{B} = b_1, b_2, ..., b_K$ specify spatial layout constraints. The grounding resampler $\Phi gr$ processes multi-scale subject features with positional encodings to create spatially-aware representations:

$$\mathbf{F}_{\text{gr}} = \Phi_{\text{gr}}\left(\text{CLIP}(\mathbf{I}_{\text{ref}}), \text{FE}(\mathbf{B})\right) \quad (2)$$

While MS-Diffusion achieves impressive layout-guided generation capabilities, our analysis reveals several critical limitations that stem from operating primarily at the pixel and feature level without explicit semantic consistency guarantees. These limitations manifest in three primary failure modes: **Subject Identity Drift:** Generated subjects gradually deviate from reference appearance as the diffusion process progresses, particularly in complex multi-subject scenarios where competing subject features can interfere with each other. **Cross-Subject Contamination:** Features from different subjects blend inappropriately, resulting in subjects that combine characteristics from multiple references rather than maintaining distinct identities. **Spatial Incoherence:** Global image structure lacks semantic consistency, leading to compositionally plausible but semantically implausible arrangements where subjects may appear in contextually inappropriate relationships or spatial configurations.

## 3.2 Problem Analysis

### 3.2.1 The semantic alignment gap in diffusion models

Current multi-subject diffusion models operate under an implicit assumption that pixel-level reconstruction fidelity directly translates to semantic consistency. This assumption, while reasonable for simple single-subject scenarios, breaks down catastrophically in complex multi-subject environments where the model must navigate a vastly expanded solution space. We identify this fundamental disconnect as the semantic alignment gap - a critical mismatch between learned diffusion representations and the semantic content they are intended to encode. This gap emerges because standard diffusion training exclusively optimizes reconstruction error in latent space. However, this objective does not provide guarantees about semantic preservation. Two images may exhibit similar pixel statistics while containing vastly different semantic content, or conversely, semantically identical images may show significant pixel-level variations due to natural variations in pose, lighting, expression, or background context. The semantic alignment gap becomes particularly pronounced in multi-subject scenarios where the model must simultaneously satisfy multiple semantic constraints. Each subject introduces its own semantic requirements, and the interaction between these constraints creates a complex optimization landscape where pixel-level optima may correspond to semantically inconsistent solutions.

### 3.2.2 Attention mechanisms and semantic correspondence challenges

Multi-subject diffusion adapters rely heavily on specialized attention mechanisms to establish correspondences between text descriptions and visual features. In multi-subject scenarios, both cross-attention and self-attention mechanisms face unique challenges that traditional training approaches fail to address adequately. The standard cross-attention computation:

$$\text{Attn}(Q, K, V) = \text{softmax}\left(\frac{QK^T}{\sqrt{d_k}}\right) V \tag{3}$$

establishes correspondences between spatial locations (queries $\mathbf{Q}$) and subject/text information (keys $\mathbf{K}$, values $\mathbf{V}$). While this mechanism can learn statistical correlations between spatial regions and subject features, it lacks explicit semantic grounding, leading to learned associations that may be statistically valid but semantically inconsistent. The adapter's self-attention mechanisms handle long-range dependencies and global structural relationships, but in multi-subject contexts, they must balance competing structural requirements from different subject while maintaining overall compositional coherence. Without semantic anchoring, self-attention may learn to prioritize pixel-level consistency over semantic plausibility. These limitations manifest as two problems: **Local semantic drift:** Cross-attention mechanisms may initially learn appropriate subject-region associations, but without continuous semantic supervision, these associations gradually drift during training, leading to progressive identity degradation. **Global Structural Incoherence:** Self-attention mechanisms optimize for local spatial consistency but lack global semantic constraints, potentially resulting in compositions that are locally coherent but globally implausible from a semantic perspective.

Previous approaches attempt to address these issues through careful loss function design, architectural modifications, or dataset curation. However, we argue that these approaches treat symptoms rather than the root cause. The fundamental issue requires semantic anchoring - explicitly grounding diffusion adapter's representations in a semantically meaningful feature space that provides stable, consistent representations of visual concepts across variations in appearance, pose, and context. DINOv2 emerges as an ideal semantic anchor due to crucial properties that make it uniquely suited for this role. DINOv2's self-supervised training on diverse visual data creates representations that capture semantic content independent of superficial visual variations, providing stable semantic anchors even when appearance varies significantly. The model captures both fine-grained object details through patch-level representations and global scene structure through holistic embeddings, enabling alignment at multiple levels of semantic abstraction.

## 3.3 Dual Alignment Framework

Our approach introduces a novel dual alignment framework that establishes explicit semantic correspondence bridges between the adapter's learned feature space and DINOv2's semantically grounded feature space. The key architectural insight driving our design is that different attention mechanisms within the adapter serve different purposes and therefore require different alignment strategies:

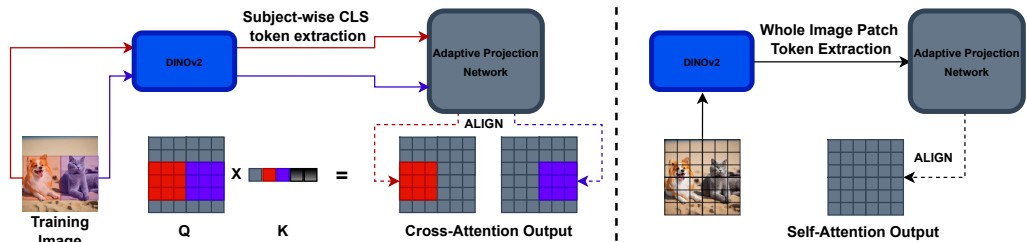

Figure 2: Detailed diagram of our novel Dual Alignment Framework. From left to right: (1) Spatially-Aligned Subject-Specific Cross-Attention. (2) Patch-Aligned Self Attention

- **Spatially-Aligned Subject-Specific Cross Attention** handles subject-specific information processing and spatial localization decisions. This mechanism determines which subject features should influence which spatial locations, making it crucial for maintaining subject identity and preventing cross-subject contamination. For this attention mechanism, we require alignment with subject-specific semantic representations that capture individual identity information
- **Patch-Aligned Self Attention** manages global structural relationships and long range spatial dependencies. This mechanism ensures overall compositional coherence and handles the integration of multiple subjects into a unified scene. For this mechanism, we require alignment with (TBF)

This architectural separation enables our framework to address both local identity preservation and global structural coherence simultaneously, providing comprehensive semantic supervision across all levels of the adapter's processing pipeline while keeping the base diffusion model frozen.

### 3.3.1 Spatially-Aligned Subject-Specific Cross Attention

The first pathway of our framework focuses on ensuring that cross-attention mechanisms maintain accurate subject-region correspondences. This involves several sophisticated components working in tandem. For each subject $j$ in a multi-subject composition, we extract spatial regions corresponding to that subject's designated location using precise bounding box information. This extraction process creates subject-specific masks $M_j$ that identify pixels belonging to each individual subject:

$$M_j(x, y) = \begin{cases} 0 & \text{if } (x, y) \in \text{BBox}_j \\ -\infty & \text{otherwise} \end{cases} \tag{4}$$

The cross-attention outputs are isolated corresponding to each spatial region, extracting features $\mathbf{h}_{\text{cross}}^{(j)}$ that represent how the adapter processes information for subject $j$. This isolation ensures that we can evaluate and guide the model's understanding of individual subjects independently. For each subject, we extract DINOv2 CLS tokens from the corresponding reference image regions. These tokens provide semantically rich, holistic representations of each subject that capture identity-relevant information while being robust to variation in pose, expression and lighting. We establish correspondence between the adapter's cross-attention features and DINOv2 CLS tokens through normalized cosine similarity:

$$L_{\text{cross}}^{(j)} = 1 - \frac{\left\| \text{Pool}\left(\mathbf{h}_{\text{cross}}^{(j)}\right) \right\| \left\| \Pi_{\text{cross}}\left(\mathbf{f}_{\text{dinov2}}^{\text{cls},(j)}\right) \right\|}{\left\langle \text{Pool}\left(\mathbf{h}_{\text{cross}}^{(j)}\right), \Pi_{\text{cross}}\left(\mathbf{f}_{\text{dinov2}}^{\text{cls},(j)}\right) \right\rangle} \tag{5}$$

This alignment ensures that the adapter's internal representation of each subject region remains semantically consistent with the actual subject identity.

### 3.3.2 Patch-Aligned Self Attention

The second pathway addresses global structural coherence by aligning self-attention mechanisms with comprehensive scene-level semantic representations. Self-attention mechanisms of the adapter

capture long-range spatial relationships and global structural patterns. We extract global features $\mathbf{h}_{\text{self}}$ without spatial masking, preserving the adapter's holistic understanding of the scene composition. Entire reference images are processed through DINOv2 to extract patch-level tokens that capture comprehensive scene structure. These patch tokens provide detailed spatial-semantic correspondence information that reflects how different regions of the image relate to each other. Finally, we align the adapter's self-attention outputs with DINOv2 patch token representations through global pooling and projection:

$$L_{\text{self}} = 1 - \frac{\left\| \text{Pool}\left(\mathbf{h}_{\text{self}}\right) \right\| \left\| \Pi_{\text{self}}\left(\text{Pool}\left(\mathbf{F}_{\text{dinov2}}^{\text{patch}}\right)\right) \right\|}{\left\langle \text{Pool}\left(\mathbf{h}_{\text{self}}\right), \Pi_{\text{self}}\left(\text{Pool}\left(\mathbf{F}_{\text{dinov2}}^{\text{patch}}\right)\right) \right\rangle} \tag{6}$$

This alignment prevents global structural inconsistencies by ensuring that the adapter's overall compositional understanding maintains coherence with the reference image structure.

### 3.3.3 Adaptive Projection Network

Rather than using fixed mappings between feature spaces, we employ trainable projection networks that adapt to the specific characteristics of different adapter layers and semantic contexts:

$$\Pi(\mathbf{f}) = \text{Linear}_2\left(\text{GELU}\left(\text{Dropout}\left(\text{LayerNorm}\left(\text{Linear}_1(\mathbf{f})\right)\right)\right)\right) \tag{7}$$

This ensures that the adapter's projection networks can learn sophisticated mappings between adapter representations and DINOv2 feature space ($\mathbb{R}^{d_{\text{adapter}}} \to \mathbb{R}^{d_{\text{dinov2}}}$) while maintaining numerical stability throughout the training process.

### 3.3.4 Multi-Scale Semantic Supervision Strategy

Different blocks within the adapter capture different levels of abstraction, from low-level visual patterns in early layers to high-level semantic concepts in deeper layers. Our framework applies alignment across multiple carefully selected adapter layers to capture semantic correspondences at different granularity. We focus alignment on Down Blocks and Mid Block within the adapter, which correspond to the most meaningful processing stages in the adapter architecture. These layers capture intermediate and high-level semantic information while avoiding over-constraining low-level visual details. We elaborate further regarding the choice of alignment blocks in section 4.5. The complete training objective seamlessly integrates diffusion denoising with adapter semantic alignment while maintaining careful balance between generation quality and semantic consistency:

$$L_{\text{total}} = L_{\text{Diffusion}} + \alpha.L_{Align} \tag{8}$$

The hyperparameter $\alpha$ controls the relative importance of representation alignment versus pixel-level reconstruction. This parameter requires careful tuning to ensure that semantic consistency is enforced within the adapter without compromising the fundamental generation capabilities of the overall system.

## 4 Experimental Studies

### 4.1 Training strategy and implementation details

Our implementation utilizes PyTorch with mixed-precision training on a single NVIDIA L40S Tensor Core GPU with 48GB VRAM. The DINOv2-Base model, with a layer depth of 12, serves as a feature extractor for our framework. Training procedure builds upon Stable Diffusion XL as our base generative model which we freeze and incorporate our adapter on top of. We follow a two-stage training curriculum that involves: **(1)** Initialize with $\alpha = 0$, training exclusively on the diffusion loss $L_{Diffusion}$ for the initial phase. This allows the adapter modules to learn fundamental image generation capabilities while preserving the pre-trained SDXL model's knowledge. **(2)** Introduce alignment loss with $\alpha = 0.01$ (empirically chosen), enabling the Dual Alignment Framework. We find that this setup allows the adapter to grasp image generation fundamentals while also aligning the generations to be more identity preserving and closer to the reference subjects.

## 4.2 Datasets and Evaluation protocol

We utilize a curated dataset of 10,000 multi-subject compositions called Subject Dataset 10k from [34]. Each image contains 1-4 subjects with corresponding bounding box annotations, textual descriptions, and segmentation masks. The dataset covers diverse categories including living entities, objects, vehicles, and scenes. We utilize two complementary benchmarks to conduct comprehensive evaluation of our method: **MSBench** A systematic benchmark framework adapted from [17], which provides a thorough assessment of compositional generation capabilities across varying subject complexities. The benchmark encompasses 9 two-subject scenarios and 4 three-subject scenarios. We generate 30 samples per combination type across different random seeds for robust analysis. **CustomConcept101** We conduct single-subject evaluation using the established CustomConcept101 benchmark from [20], which provides a standardized framework for asserting single-subject generation fidelity, identity preservation and prompt adherence across diverse prompt categories. The evaluation encompasses 101 carefully curated subjects, where each subject contains 3-15 reference images. For each reference image within every subject, we generate one corresponding output image using different random seeds to ensure stochastic diversity across the evaluation set. This approach yields a total of 634 generated samples across all 101 subjects, providing comprehensive coverage of generation performance across the full dataset. Our evaluation protocol employs three metrics that provide comprehensive assessment of model performance: **CLIP-I** Measures visual similarity between generated and reference images using CLIP's visual encoder. **CLIP-T** Evaluates semantic alignment between generated images and text prompts using CLIP's joint vision-language embedding space. **DINO** Leverages DINOv2's robust self-supervised representations to evaluate semantic consistency.

Table 1: Performance comparison of different models on single and multi-subject tasks

| Model | Single Subject | | | Multi Subject | | |
|---|---|---|---|---|---|---|
| | CLIP-I | CLIP-T | DINO | CLIP-I | CLIP-T | DINO |
| $\lambda$-Eclipse | 0.8240 | 0.2843 | 0.6833 | 0.7083 | 0.3035 | 0.3909 |
| OmniGen | 0.8060 | 0.2931 | 0.6530 | 0.7088 | 0.2814 | 0.4539 |
| MS-Diffusion | 0.8172 | 0.2898 | 0.6845 | 0.7115 | **0.3302** | 0.4648 |
| **MultiPersona-Align (Ours)** | | | | | | |
| **Cross-Attention Alignment Only** | 0.8203 | 0.2912 | 0.6887 | 0.7201 | 0.3281 | 0.4682 |
| **Cross+Self Attention Alignment** | **0.8266** | **0.2960** | **0.6925** | **0.7275** | 0.3297 | **0.4758** |

## 4.3 Quantitative Results and Analysis

We compare against three baselines that span different paradigms in multi-subject generation: MS-Diffusion as the current state-of-the-art in spatial masking approaches, $\lambda$-Eclipse for layout-aware conditioning methods, and OmniGen representing unified multi-modal generation frameworks. These baselines collectively represent leading approaches to zero-shot multi-subject personalized image synthesis, enabling comprehensive evaluation across different technical strategies.Table 1 presents results comparing our method against state-of-the-art baselines across single and multi-subject generation tasks. In single-subject evaluation on CustomConcept101, our method achieves competitive performance with slight improvements over baselines: CLIP-I score of 0.8266 vs 0.8240 for $\lambda$-Eclipse, indicating superior visual fidelity preservation. The CLIP-T score of 0.2960 demonstrates effective text alignment, while our DINO score of 0.6925 reflects strong semantic consistency. These results validate that Dual Alignment Framework maintains single-subject generation quality while preparing for multi-subject extensions. The true strength of our approach emerges in multi-subject scenarios on MSBench, where we observe solid performance gains. Our method achieves a CLIP-I score of 0.7275, significantly outperforming all baselines. This average 2.2% improvement represents meaningful advancement in a challenging domain where small gains are significant. The DINO score improvement of 0.4758 vs. MS-Diffusion's 0.4648 demonstrates our framework's effectiveness in preserving semantic consistency across multiple subjects. Further details are provided in section 4.5

### 4.4 Qualitative Analysis

Figure 3 presents qualitative comparisions demonstrating our method's advantages across diverse multi-subject scenarios. The visual results reveal several key improvements over existing approaches which we will go over row-wise:

- **Row 1:** Our method successfully preserves the dog's breed-specific characteristics including ear shape, coat texture, and facial structure, while simultaneously maintaining the cat's distinctive striped pattern and facial features. In contrast, MS-Diffusion exhibits subject identity drift where the cat's eyes and fur becomes less preserved, and OmniGen loses photorealism and shows hallmarks of an AI-generated image though it does capture essential subject characteristics. $\lambda$-eclipse struggles with fine details like the eyes and fur, and the overall lighting of the scene is unnatural.

- **Row 2:** This indoor composition presents unique challenges in differentiating between a living subject and an inanimate object. $\lambda$-eclipse fails at this differentiation and generates an image that exhibits texture and feature bleeding between the subjects. MS-Diffusion tends to over-smooth the plushie and misses the eyes of the cat, while OmniGen struggles to produce a coherent generation of the plushie. Our approach demonstrates superior performance in preserving the distinct textural properties of each subject.

- **Row 3:** Our method excels in preserving the details of both subjects and naturally rendering them in the sunny meadow with appropriate lighting. Baseline methods seem to struggle with generating a plausible image with regards to subject placement, lighting, and coherence with the environment

- **Row 4:** This composition challenges the model with subjects having distinctly different material properties and textural characteristics. Our approach demonstrates exceptional material differentiation, preserving the boot's leather texture and stitching details, while maintaining the teddybear's soft, fabric appearance and characteristic stuffed posture. The wooden floor context is rendered with appropriate grain patterns and realistic lighting.

Comparative analysis reveals significant baseline limitations: MS-Diffusion misses finer details and the separation between the subjects. OmniGen fails to achieve convincing material rendering, with subjects appearing flat and lacking the three-dimensional quality necessary for believable object representation, while $\lambda$-eclipse exhibits significant subject bleeding.

### 4.5 Ablation Studies

**Dual Alignment**   We conduct systematic ablation studies examining the effect of each alignment pathway. Implementing alignment only in the cross-attention mechanism (section 3.3.1) yields CLIP-I: 0.7201, CLIP-T: 0.3281, DINO: 0.4682 on multi-subject tasks. This configuration improves subject identity preservation compared to baselines but lacks global coherence. To ensure that our subjects and background are coherent with lighting and spatial composition, we add our alignment to the self-attention mechanism as well (section 3.3.2) and the results are shown in table 1.

**Alignment Mappings** The selection of optimal alignment mappings between DINOv2 layers and adapter blocks is a critical design decision that directly impacts generation quality. Our empirical analysis reveals that the hierarchical nature of U-Net architectures mandates specific constraints on effective alignment strategies. DINOv2's transformer layers exhibit a progressive feature hierarchy from low-level visual patterns to high-level semantic concepts. Similarly, U-Net's Down Blocks extract increasingly abstract features, the Mid Block operates at the highest level of abstraction, containing crucial information about subject identity and Up Blocks reconstruct spatial details from compressed representations. A critical observation from our experiments is that aligning DINOv2 features with Up Blocks consistently degrades image fidelity because Up Blocks are designed to reconstruct spatial details from the abstract representations provided by the Mid Block. Introducing external alignment supervision at this stage disrupts carefully learned reconstruction pathways. Therefore we restrict our alignment exclusively to Down Blocks and Mid Block. After extensive systematic experimentation, we identify the following alignment strategy to be optimal: (1) Aligning each subject's region of the cross-attention output with the final layer CLS features of those subject regions in the reference image, (2) Aligning the self-attention outputs with the final layer patch features of the reference image.

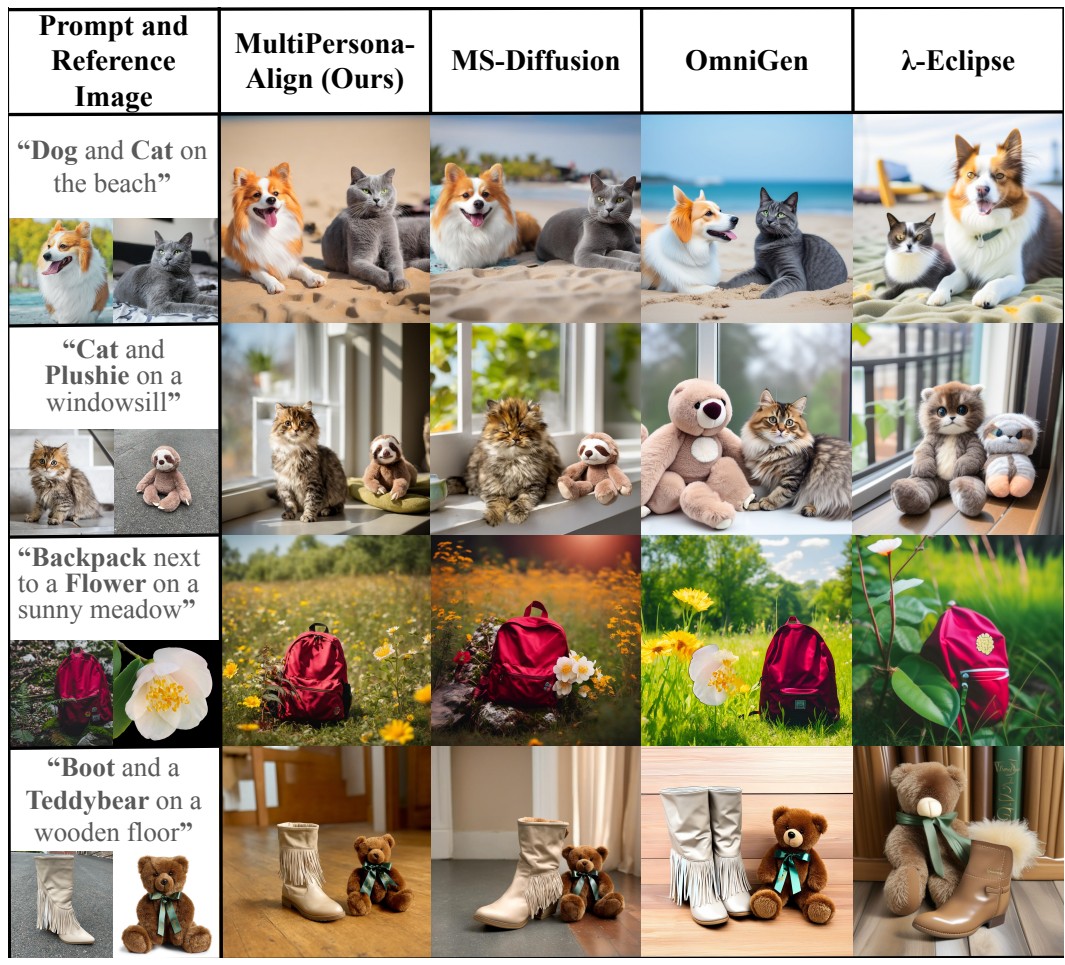

Figure 3: Qualitative comparison of different models

## 5 Discussion and Conclusion

We introduced MultiPersona-Align, a novel approach for zero-shot multi-subject personalized image generation that addresses fundamental limitations in existing diffusion-based methods. Our Dual Alignment Framework systematically tackles both local subject consistency and global image coherence through complementary mechanisms. This approach operates without requiring additional training data, architectural modifications to base models, or subject-specific fine-tuning. Extensive experiments demonstrate consistent improvements across standard benchmarks, with particularly notable gains in challenging multi-subject scenarios where existing methods struggle most, while introducing minimal computational overhead and maintaining inference efficiency. While our framework demonstrates significant improvements in multi-subject personalized generation, several limitations suggest directions for future investigation. Although our method introduces minimal computational overhead during training (15% with Dual Alignment) and no overhead during inference, future work could explore more efficient alignment strategies. Our current framework focuses on visual-semantic alignment. Incorporating linguistic semantic alignment beyond CLIP could further improve text adherence in complex multi-subject scenarios. Adaptive alignment scheduling based on training progress could optimize the balance between generation quality and semantic consistency. While our evaluation covers diverse scenarios, expanding to additional domains (artistic styles, specialized objects, cultural contexts) would provide broader validation of the approach's generalizability. The success of our alignment-based approach suggests that explicit semantic supervision represents a promising direction for enhancing personalized generation capabilities. By bridging the gap between pixel-level reconstruction objectives and semantic consistency requirements, we provide a foundation for more reliable and controllable multi-subject image synthesis.

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
