# OpenReview forum: "MultiPersona-Align: Zero-Shot Multi-Subject Personalized Image Generation with Layout-Guidance via Dual Representation Alignment"
_NeurIPS.cc/2025/Workshop/UniReps — UniReps2025_

### Official Review · Reviewer_H7j7 · 2025-09-12
**Well done, exceptional work! A few organizational changes would make it an outstanding paper.**

**Confidence:** 4

**Review:**

Paper: MultiPersona-Align: Zero-Shot Multi-Subject Personalized Image Generation with Layout-Guidance via Dual Representation Alignment

Summary:

The paper proposes a framework and set of architectures to solve the problem of image generation when there are multiple subjects in a generated image. The challenge the author’s claim to address is the difficult balance between image fidelity and semantic meaning, when multiple subjects are being represented in a single image.

They implement a framework consisting of existing work using the grounding resampler (from MS-Diffusion), identify its shortcomings for multi-subject image generation, and attempt to solve them with a “Dual alignment framework”. It adds cross-attention and self attention feature alignment to the image generation process, helping combat issues such as meshing of features across subjects, changing subject locations in the image, or altering the forms of the object in the reference image. These problems can be seen in the various State-of-the-art output images within the qualitative analysis of the paper.

Quality:

The quality of the work overall is very good. The justifications of architecture changes and additions are all well explained, with well structured and clear diagrams. Every component is well formulated. You also provide ample details, dataset information, and results for your work.

Clarity:

This was a difficult paper to parse. The ordering of your background leading into your claimed contributions, as well as the order in which you explain your framework can be confusing to follow. Each section made sense within itself, and was well written. However I highly recommend finding a more intuitive re-ordering of your explanations throughout the paper to increase clarity.

Significance:

Quantitative gains were hard to understand until further context later in the paper, they appear like minimal improvements. However, (as stated by the author) these gains are significant for the context of this problem. That was not clear until Qualitative analysis put those improvements into context, which show how and why those seemingly small improvements over the state-of-the-art are significant. Ultimately, this does show a clearly significant improvement over other models’ performance.

You frequently mention DINO, is this purely a solution for DINOv2? Is it adaptable to other foundation models and a model agnostic improvement? Or are there certain constraints a model must satisfy to work with your solution? I don’t believe these are addressed in the paper, but they would be important to mention at some point, even if only in your conclusions.

Originality:

If the claims and citations of the authors are correct (which I do believe are correct) this is a novel solution to address a common problem within the domain of image generation. The authors do a good job identifying the problem (the current trade-off between semantic representations and image quality when multiple subjects are generated together) and a potential solution (creating a framework such as theirs highlights the potential need for an addition of various semantic extraction and attention mechanisms to better generate images with multiple subjects).

Additional note: I would suggest making it clearer that you are not proposing any portion of an existing MS-Diffusion as part of your contributions (such as the grounding resampler). It would better highlight your actual contributions and improve clarity when looking at what your paper is contributing. This mainly caused confusion because your “Background” section is in your “Proposed work” section. Perhaps consider separating these.

**Score:**

4

**Topic Fit:**

3

---

### Official Review · Reviewer_GJV8 · 2025-09-13
**Comment**

**Confidence:** 3

**Review:**

This paper addresses the challenging problem of multi-subject personalized image generation with spatial control and semantic consistency. The proposed Dual Alignment Framework leverages DINOv2 features to improve both subject-specific fidelity and global coherence. The experiments are extensive, covering multiple benchmarks, with quantitative and qualitative results that show consistent improvements over strong baselines.

I would recommend acceptance without hesitation if this were submitted as an extended abstract, since the problem is important and the approach is cleanly presented. However, as a full proceedings paper, I feel the work still has several weaknesses and unclear aspects that should be further improved:

(1) The proposed dual alignment may be perceived as an incremental extension of existing representation alignment methods. It is not entirely clear what is fundamentally novel beyond substituting CLIP with DINOv2 and applying alignment at both cross- and self-attention levels.

(2) While results are consistent, the reported gains are relatively small in magnitude.

(3) The need of CLIP-based alignment baselines.

**Score:**

3

**Topic Fit:**

3

---

### Official Review · Reviewer_qAnT · 2025-09-15
**The work addresses genuine problems in multi-subject generation but lacks theoretical insights into representation/neural learning.**

**Confidence:** 4

**Review:**

This paper presents MultiPersona-Align, a dual alignment framework for multi-subject diffusion models that addresses semantic consistency issues through DINOv2 representation alignment. The technical approach systematically separates cross-attention alignment using subject-specific CLS tokens from self-attention alignment using global patch representations.

The dual alignment framework maintains compatibility with existing architectures like Stable Diffusion XL while requiring only 15% additional training overhead. It is hard to systematically compare qualitative results, but they seem to show effective preservation of subject-specific characteristics such as breed features in animals and material properties in objects while maintaining spatial coherence.

However, the paper lacks on several fronts: The computational analysis lacks depth, mentioning training overhead without detailed timing or memory profiling during inference. The theoretical justification for architectural choices remains superficial, relying primarily on empirical validation rather than principled design decisions. Any inspiration from neuroscience principles in designing the architecture?

The work addresses genuine problems in multi-subject generation where spatial masking approaches fail to provide semantic consistency guarantees. However, the conceptual novelty remains incremental, building primarily on established techniques in attention mechanisms and representation alignment rather than introducing fundamental algorithmic innovations. The integration of DINOv2 as a semantic anchor constitutes an application of existing self-supervised representations rather than theoretical insights into representation/neural learning.

**Score:**

2

**Topic Fit:**

2